# Compact zinc finger architecture utilizing toxin-derived cytidine deaminases for highly efficient base editing in human cells

Friedrich Fauser[1], Bhakti N. Kadam[1], Sebastian Arangundy-Franklin[1], Jessica E. Davis[1], Vishvesha Vaidya[1], Nicola J. Schmidt[1], Garrett Lew[1], Danny F. Xia [1], Rakshaa Mureli[1], Colman Ng[1], Yuanyue Zhou[1], Nicholas A. Scarlott [1], Jason Eshleman[1], Yuri R. Bendaña[1], David A. Shivak[1], Andreas Reik[1], Patrick Li[1], Gregory D. Davis[1] & Jeffrey C. Miller [1] ✉

Nucleobase editors represent an emerging technology that enables precise single-base edits to the genomes of eukaryotic cells. Most nucleobase editors use deaminase domains that act upon single-stranded DNA and require RNA-guided proteins such as Cas9 to unwind the DNA prior to editing. However, the most recent class of base editors utilizes a deaminase domain, $DddA_{tox}$, that can act upon double-stranded DNA. Here, we target $DddA_{tox}$ fragments and a FokI-based nickase to the human CIITA gene by fusing these domains to arrays of engineered zinc fingers (ZFs). We also identify a broad variety of Toxin-Derived Deaminases (TDDs) orthologous to $DddA_{tox}$ that allow us to fine-tune properties such as targeting density and specificity. TDD-derived ZF base editors enable up to 73% base editing in T cells with good cell viability and favorable specificity.

Nucleobase editors enable the targeted deamination of either a cytosine or an adenine base within the nuclear genome of eukaryotic cells[1–3]. As such they are part of the molecular toolbox of genome editing reagents, including engineered nucleases[4–7], prime editors[8], and site-directed recombinases[9]. Cytosine base editors (CBEs) mediate a targeted C•G-to-T•A base pair change and are particularly useful for applications that involve knocking out the expression of a gene through the introduction of a nonsense mutation. CBEs are created by fusing a cytidine deaminase to a DNA binding domain and require nearby DNA to be nicked to achieve optimal activity[3]. CBEs can also contain a uracil glycosylase inhibitor (UGI) to inhibit base excision repair that would otherwise revert the edited base back to cytosine. Most CBEs use an RNA-guided CRISPR/Cas protein such as Cas9 to target the cytidine deaminase to the desired genomic locus, optionally nick the non-deaminated DNA strand, and unwind the DNA to provide a single-stranded DNA substrate in an R-loop for the fused cytidine deaminase. First generation Cas9 CBEs supported base editing efficiencies between 15 and 75%[3], but the size of the entire expression cassette precludes packaging in a single AAV vector[10], and delivery is

also limited to the nuclear genome due to constraints on sgRNA delivery to organelles[11,12]. It is possible to package these constructs into two separate AAV vectors using a recently described trans-splicing intein architecture[10], but this would presumably be a more challenging and less efficient system compared to a base editor that could be delivered using a single AAV vector. More compact CBEs consisting of ZFs fused to APOBEC or AID deaminases have been reported, but these had lower efficiencies[13] likely due in part to the inability of the ZFs to unwind the DNA to create good substrates for these types of deaminases.

More recently, a class of cytidine deaminases that do not require CRISPR/Cas-mediated DNA unwinding for effective base editing has been reported, namely the toxin-conferring deaminase domain from the *Burkholderia cenocepacia* double-stranded DNA cytidine deaminase toxin A ($DddA_{tox}$)[12]. In addition to mitochondrial DNA editing, the authors were able to demonstrate up to 27% base editing in the nucleus of human cells by splitting $DddA_{tox}$ into two fragments that are inactive on their own and then fusing these complementary fragments to separate engineered transcription activator-like effector (TALE)

[1]Sangamo Therapeutics, Inc., Brisbane, CA, USA. ✉e-mail: jmiller@sangamo.com

domains. Since then, other groups have utilized these DddA$_{tox}$ fragments fused to both TALEs and ZFs with a variety of delivery modalities to edit the mammalian mitochondrial genome[14–21], the mammalian nuclear genome[17,22,23], the mitochondrial genome of other model systems such as zebrafish[19,24], as well as the plant mitochondrial[25] and chloroplast genomes[25,26]. For example, mitochondrial base editing has been demonstrated for TALE-targeted CBEs (TALE-CBEs) via in vivo mRNA delivery in rats[18] and mice[15], in vitro mRNA delivery in mouse[16] and human[14,21] embryos, and through AAV delivery in post-mitotic mouse tissue[20]. Nuclear and mitochondrial DNA editing was also demonstrated in human cells with ZF-targeted CBEs (ZF-CBEs) and base editing frequencies of up to 60% were achieved within the nuclear genome[17,23]. Boyne et al.[22] recently reported up to 86% base editing in T cells using TALE-targeted CBEs[22]. Higher editing efficiencies might have been obtained if the authors had also included a nickase to nick the non-deaminated strand[17], but the added size of a Cas9 nickase[5], a TALE nickase[27] or an independent ZF nickase[28,29] would preclude AAV packaging in a single vector. One solution to address AAV packaging constraints is the development of non-toxic, full-length DddA$_{tox}$ variants to make smaller monomeric DddA-derived CBEs[30]. Notably, TALE-derived nickases based on MutH or Nt.BspD61(C) were recently explored for more efficient mitochondrial base editing[31]. In addition, catalytically inactive DddA$_{tox}$ was successfully used to provide a single-stranded DNA substrate for an adenine deaminase, supporting the development of compact protein-guided adenine base editors for targeted A•T-to-G•C base editing[32].

In this study, we build upon the work of Mok et al.[12] by first replacing the TALE DNA-binding domain with arrays of engineered ZFs. Each 34-residue TALE repeat recognizes a single basepair of target DNA, whereas each 28-residue ZF recognizes three basepairs allowing for a much smaller construct containing fewer repeated domains to target the same DNA sequence. In order to increase the editing efficiency, we also include a ZF nickase[28,29] which is a dimeric ZF nuclease where one of the FokI cleavage domains is catalytically inactive. In an effort to keep the overall construct small enough for AAV packaging, we fuse one deaminase fragment and one copy of the FokI DNA cleavage domain to opposite ends of the same ZF array, resulting in a three-peptide CBE-nickase system. In our initial characterizations, we confirm that constructs containing the DddA$_{tox}$ deaminase domain fragments employed by Mok et al.[12] are limited in terms of what types of sequences they can target with a strong 5′ thymidine preference adjacent to the target cytosine (5′-TC). Mok et al.[12] also observed

editing at 5′-TCC motifs as well as minor activity levels at some 5′-AC motifs for the canonical DddA$_{tox}$, and later increased the targeting scope of DddA$_{tox}$ through phage-assisted non-continuous and continuous evolution[33]. We choose a different strategy to further expand the base editing properties of ZF-CBEs by exploring deaminase domains derived from other interbacterial toxins. A similar effort recently described a DddA homolog from *Simiaoa sunii* (Ddd_Ss) which can efficiently deaminate cytosine in 5′-AC, GC, and TC contexts[34]. Here, we identify more than 10 Toxin-Derived Deaminase (TDD) domains with high activity in human cells and a relaxed preference for the base adjacent to the target cytosine. We also identify a TDD domain from *Pseudoduganella violaceinigra* (TDD14) that enables up to 73% base editing in human T cells with minimal indels, good cell viability and better genome-wide specificity compared to DddA$_{tox}$-derived reagents. The edited cells demonstrate the expected gene knockout phenotype and the entire ZF-CBE-nickase system can meet the construct size requirements for packaging in a single AAV vector, thus showing promise for potential future therapeutic applications in vivo.

## Results

### Development of ZF-CBE-nickases

We first wanted to determine if engineered ZF-DddA$_{tox}$ fusions can achieve similar activity levels of nuclear base editing as TALE-DddA$_{tox}$ fusions. Thus, we targeted a region within the human CCR5 gene (Supplementary Fig. 1a) that was previously edited with TALE-CBEs[12]. Our dimeric ZF-CBE constructs utilized the same DddA$_{tox}$-G1333 split variant and were active in human K562 cells (Supplementary Fig. 1b and Supplementary Data 1), although editing levels were below that reported by Mok et al.[12] for their TALE-CBEs. We then designed ZF nickases[28,29] to nick the non-deaminated strand and therefore bias the cells' DNA mismatch repair machinery to favor the intended editing outcome. All FokI nickases described in this study are based on heterodimeric ELD and KKR FokI variants[35] to avoid the assembly of an active nuclease through homodimerization at potential off-target sites. To create a ZF-CBE-nickase, we fused a copy of the FokI cleavage domain to the N-terminus of the right ZF-CBE construct ("ZF-CBE-R" in Fig. 1A) and fused a second copy of the FokI cleavage domain containing the D450N mutation[29] to a third ZF array ("Nickase ZF" in Fig. 1A and Supplementary Fig. 1c). This ZF-CBE-nickase architecture allowed us to dramatically increase the base editing efficiency from $3.61 \pm 0.45\%$ (mean ± s.d.) for DddA$_{tox}$-G1333 without nicking to

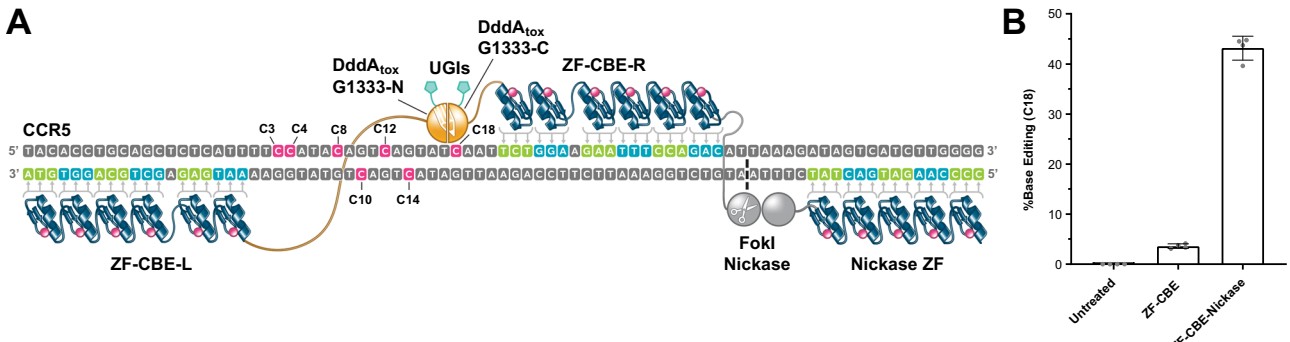

**Fig. 1 | Development of a compact ZF-CBE-nickase architecture that increases base editing efficiency. A** Sketch of a ZF-CBE-nickase used for these studies bound to its target site within the human CCR5 locus. ZF-CBE-L and ZF-CBE-R represent the left and right base editing monomers containing complementary deaminase fragments. DddA$_{tox}$-G1333-N and DddA$_{tox}$-G1333-C indicate the corresponding DddA$_{tox}$ split variants using the nomenclature from Mok et al.[12]. Each base editing monomer also contains a uracil glycosylase inhibitor (UGI) protein fused to the C-terminus of the DddA$_{tox}$ fragments. Nicking functionality can be added to this ZF-CBE by fusing one copy of the FokI cleavage domain to the N-terminus of ZF-CBE-R and fusing a second copy of the FokI cleavage domain bearing the D450N mutation to a third ZF array, depicted as Nickase ZF. Note that both ZF-CBE-L and ZF-CBE-R use a longer base skipping linker[43] between their fourth and fifth fingers (counting from the N-terminus). **B** Activity comparison of the ZF-CBE constructs shown in (**A**) with and without nicking at the most active position (C18). Data are presented as the mean ± s.d. from four biological replicates. For plotted data values, see Supplementary Data 2. Source data are provided as a Source Data file.

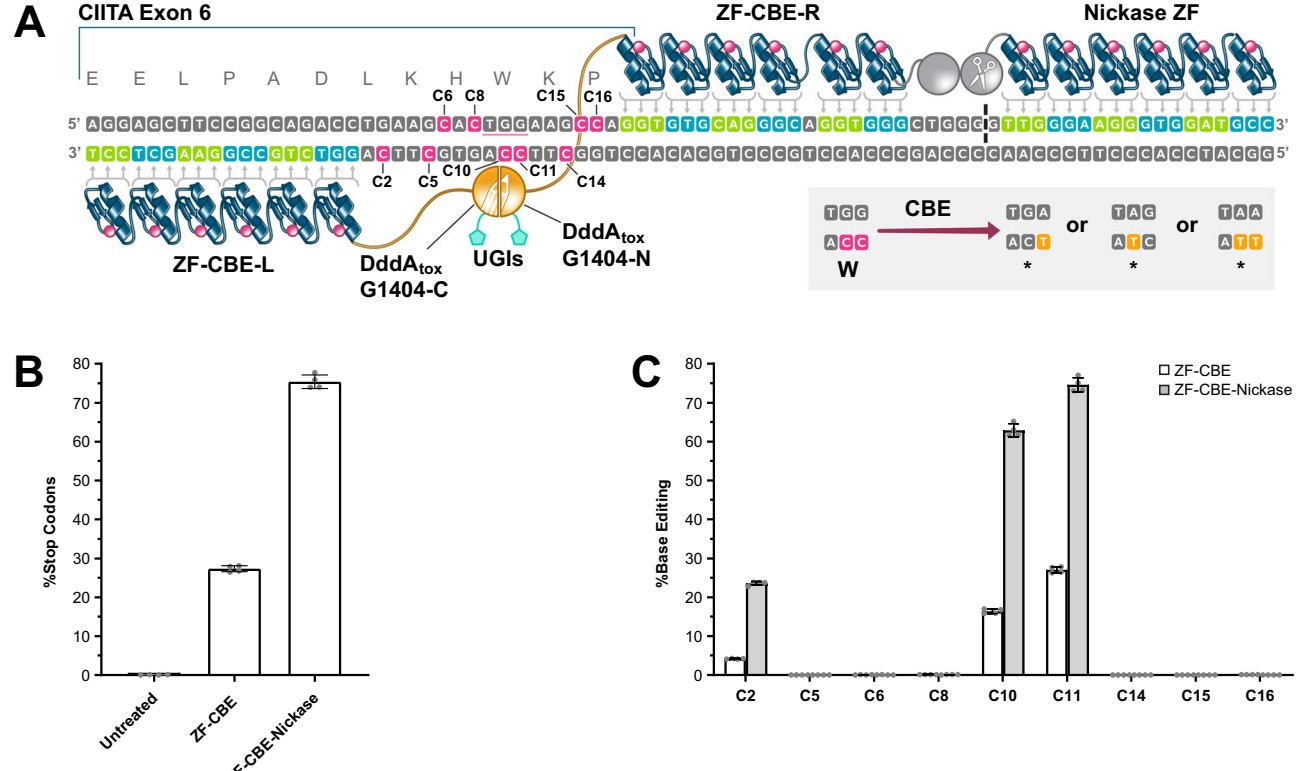

**Fig. 2 | Development of a ZF-CBE-nickase for efficient site-specific induction of stop codons within the CIITA open reading frame. A** Sketch of a ZF-CBE-nickase bound to its target site (hg38 Chr16: 10,901,523–10,901,600) within the human CIITA locus. A CBE can be used to edit the underlined tryptophan codon (TGG) and create one of three pre-mature stop codons (TGA, TAG or TAA). ZF-CBE-L binds upstream of the target codon and is fused to the C-terminal fragment of DddA$_{tox}$-G1404 while ZF-CBE-R binds downstream of the target codon and is fused to the N-terminal fragment of DddA$_{tox}$-G1404. Together, they each contain one UGI copy and form a functional ZF-CBE. Nickase functionality can be added by fusing one copy of the FokI cleavage domain bearing the D450N mutation to the N-terminus of ZF-CBE-R and fusing a copy of the FokI cleavage domain to a third ZF array,

depicted as Nickase ZF. Note that ZF-CBE-R uses a longer base skipping linker[43] between the second and third finger (counting from the N-terminus). **B** Activity comparison of the ZF-CBE constructs shown in (**A**) with and without nicking. Data are presented as the mean ± s.d. from four biological replicates. For plotted data values, see Supplementary Data 5. **C** Activity comparison of the ZF-CBE constructs shown in (**A**) across the entire base editing window. The graph shows both stop codon-inducing editing events at positions C10 and C11, as well as an unintended editing events at position C2. Data are presented as the mean ± s.d. from four biological replicates. For plotted data values, see Supplementary Data 5. Source data are provided as a Source Data file.

43.17 ± 2.40% for DddA$_{tox}$-G1333 with nicking (Fig. 1B, Supplementary Fig. 1d and Supplementary Data 2). We also explored other DddA$_{tox}$ split variants, including G1397[12] as well as G1404 and G1407 (Supplementary Fig. 1e), and noticed a considerable improvement compared to G1333 for all three variations (supplementary Fig. 1f and Supplementary Data 3). In summary, these results establish two active DddA$_{tox}$ split variants, and introduce a compact ZF-CBE-nickase architecture for more effective base editing.

### Targeting ZF-CBE-nickases to CIITA

We next wanted to investigate if ZF-CBEs were able to knock out expression of the human Class II transactivator (CIITA) for applications involving the generation of allogeneic CAR T cells[36]. We utilized the DddA$_{tox}$-G1404 split variant and designed corresponding ZF-CBEs for a site within the CIITA gene where the targeted cytosine had the required 5′-TC/5′-TCC context[12] and editing of the TGG tryptophan codon (W158) would create a pre-mature TGA, TAG, or TAA stop codon within the CIITA open reading frame (Fig. 2A). By screening a grid of eight left and eight right ZF-DddA$_{tox}$-G1404 fusion constructs (Supplementary Fig. 2a, b and Supplementary Data 4) and then adding nicking functionality (Fig. 2A and Supplementary Fig. 2c, d), we achieved up to 75.38 ± 1.76% base editing that yielded the desired stop codons in human K562 cells (Fig. 2B) with only 0.73 ± 0.07% indels (Supplementary Data 5). We noticed two limitations of ZF-CBEs

utilizing the DddA$_{tox}$-G1404 split variant to target CIITA: First, we identified 32 suitable target codons (20x CAG, 6x TGG, 4x CAA and 2x CGA) within a desired region of the CIITA gene (encoding Q21 to Q466) but only five codons provided the required 5′-TC (Q143, W158 and Q424) or 5′-TCC (Q103 and Q228) sequence context. Second, we also observed base editing at an unintended cytosine (C2) reaching up to 4.19 ± 0.08% for our most active CIITA ZF-CBE reagents without nicking, and 23.55 ± 0.49% respectively when co-delivering a nickase (Fig. 2C). In summary, these results demonstrate that ZF-CBE-nickases can be targeted beyond CCR5 to induce stop codons at a chromosomal target (CIITA) with high efficiency and minimal indels. Our results also suggest that engineered DddA$_{tox}$ variants or alternative dsDNA deaminase domains with substrate preferences beyond 5′-TC are needed to increase the targeting density of ZF or TALE-targeted CBEs.

### Alternative TDDs for dsDNA base editing

We then surveyed other TDDs in our ZF base editors in the hope of finding DddA$_{tox}$ alternatives with increased targeting density and less activity at unintended cytosines near the intended target. Our search for additional TDDs considered multiple factors including the presence at the C-terminus of a large interbacterial toxin protein and the presence of a sequence with homology to the inhibitor domain DddI in the genome of the host organism. We used the DddA$_{tox}$ sequence as a search string in both BLAST and the metagenomics database at EMBL.

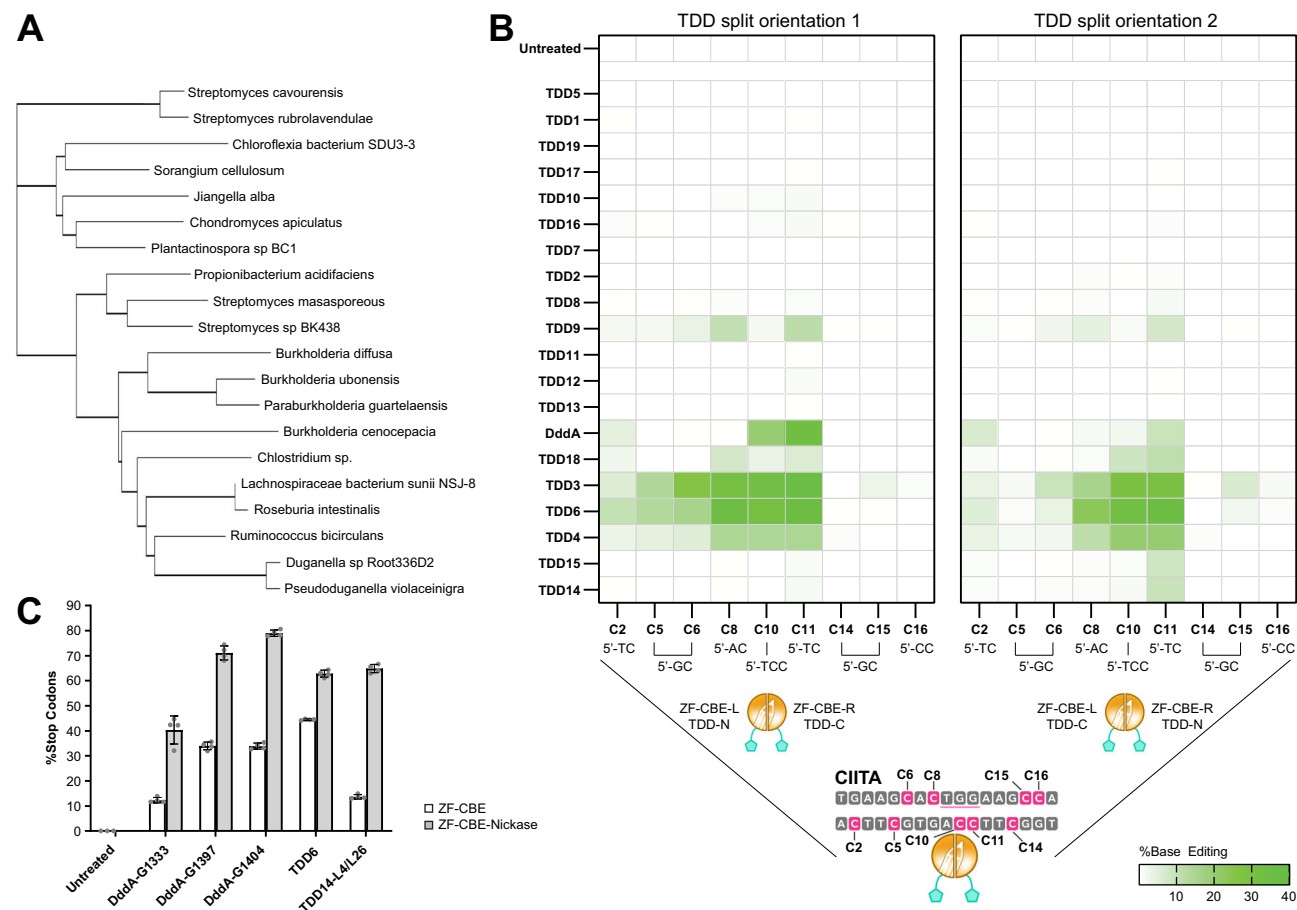

**Fig. 3 | Identification and characterization of alternative dsDNA deaminase domains for ZF-CBEs. A** Phylogenetic tree representing the sequence similarity between DddA$_{tox}$ and TDD1 to TDD19. The phylogenetic tree is labeled with the species harboring the TDD while the name of the relevant TDD is shown on the left of (**B**). Note that split positions for depicted TDDs are shown in Supplementary Fig. 3. **B** Activity comparison of ZF-TDD-derived CBEs at the CIITA locus (see Fig. 2A for reference). For each TDD up to three different split variants were tested in both orientations and the most active split variant is shown. Data are presented as the mean from four biological replicates. For additional details, see Supplementary Fig. 3c. For the full dataset and plotted data values, see Supplementary Data 6 (for

TDD1 to TDD19), and Supplementary Data 10 (for DddA$_{tox}$-G1404). Note that data for TDD20 to TDD31 is shown in Supplementary Fig. 3. **C** Activity comparison of selected TDDs from (**B**) and Supplementary Fig. 5 with and without nicking. DddA: For all three split variants, C-terminal fragment fused to ZF-CBE-L, N-terminal fragment fused to ZF-CBE-R; TDD14-G43: C-terminal fragment fused to ZF-CBE-L with linker L4, N-terminal fragment fused to ZF-CBE-R with linker L26; TDD6-R2385: N-terminal fragment fused to ZF-CBE-L with linker L26, C-terminal fragment fused to ZF-CBE-R with linker L26. Data are presented as the mean ± s.d. from four biological replicates (three for untreated sample). For the full dataset and plotted data values, see Supplementary Data 10. Source data are provided as a Source Data file.

After filtering for redundant sequences, we chose to test sequences with a high degree of similarity to the active site residues of DddA$_{tox}$. We noticed that some sequences with lower degree of similarity contained a longer peptide between the two active site cysteines, and these tended to form their own clade. We identified 30 putative deaminase domains from these two separate clades (Fig. 3A and Supplementary Fig. 3a) and tested different split variants of each TDD against the previously established CIITA target site. We first explored up to three different split variants of TDD1 to TDD19 and observed varying levels of activity including several with moderate efficiency even in the absence of a nickase, such as for example TDD3-N94 (36.95 ± 5.32% stop codons; TDD3 residues 1–94 as the N-terminal fragment and TDD3 residues 95–125 as the C-terminal fragment with the split-point after residue N94; see Supplementary Data 16 for details), TDD4-A229 (20.36 ± 5.46% stop codons; using a similar notation to denote split location) and TDD6-R2385 (TDD6 hereafter) (35.49 ± 4.27% stop codons) (Fig. 3B, Supplementary Fig. 3c and Supplementary Data 6). We later tested TDD20 to TDD30, as well as a consensus designed sequence (TDD31), with similar results, yielding additional active TDDs including TDD21, TDD25 and TDD27 (Supplementary Fig. 3b and Supplementary Data 7). Notably, we identified TDDs with relaxed 5′-TC

requirements, such as TDD6 that showed substantial base editing at 5′-GC (C5 and C6) and 5′-AC (C8) motifs within the CIITA base editing window compared to DddA$_{tox}$-G1404 (Fig. 3B). We also identified TDDs with a narrower base editing window than DddA$_{tox}$-G1404, including TDD14-G43 (TDD14 hereafter) that showed a very strong preference for position C11 (5′-TC) at the tested CIITA site (Fig. 3B). However, we noticed that TDD14 can also edit 5′-GC and 5′-AC motifs when targeted to other sites within the CIITA locus (Supplementary Fig. 4 and Supplementary Data 8). This suggests that the narrow base editing window of TDD14 is due to some other factor than just a more stringent 5′-TC motif requirement. In addition, with TDD24 we noticed editing at a base presumably bound by one of the ZF arrays at position C0 (Supplementary Fig. 3b), which was not a result observed at high frequency with other TDDs. We also explored the effect of different ZF-TDD linkers for the active TDD14 split orientation (Supplementary Fig. 5a and Supplementary Data 9), as well as both split orientations of TDD6 (Supplementary Fig. 5b, c). Our data suggests that linker optimization studies and testing both split orientations (fusing the N-terminal fragment to the left ZF-CBE arm and the C-terminal TDD fragment to the right ZF-CBE arm, or vice versa) can both be effective options for fine-tuning the base editing

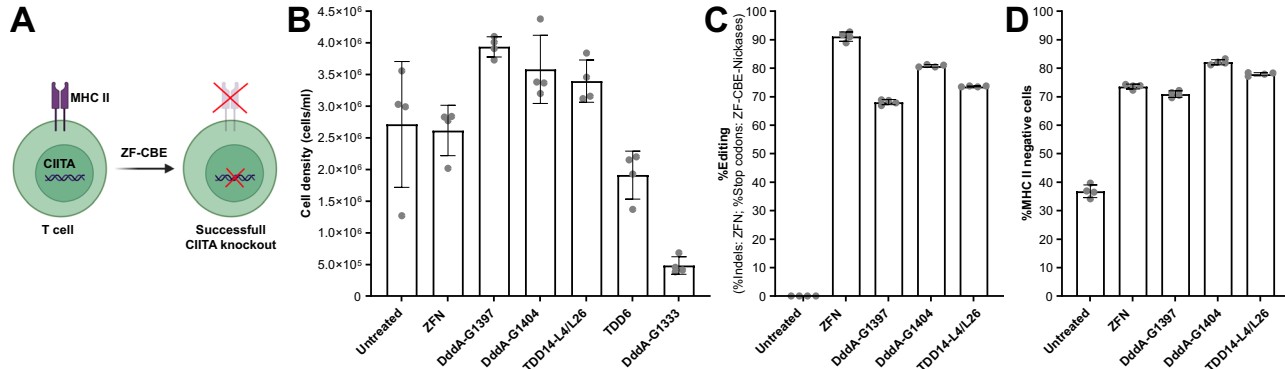

**Fig. 4 | Knocking out CIITA in human T cells using ZF-CBE-nickases. A** Sketch of a human T cell where CIITA is knocked out using a ZF-CBE-nickase. **A** was created with BioRender.com. **B** Cell density at day 10 for ZF-CBE-nickases bearing the indicated deaminase domain compared with a ZFN positive control. **C** %Stop codons for samples in (**B**) that showed acceptable cell density. **D** Percentage of MHC II negative cells for samples in (**C**). Data are presented as the mean ± s.d. from four biological replicates. For plotted data values in (**B**–**D**), see Supplementary Data 11–13. Note that Supplementary Fig. 7 shows the activity comparison of ZF-CBEs and ZF-CBE-nickases for all constructs. Source data are provided as a Source Data file.

profile within the base editing window. We recommend starting ZF-CBE designs with the L26 linker and exploring the effect of shorter linkers if necessary. Next, we added nickase functionality to selected ZF-CBEs and achieved editing levels of up to 62.80 ± 1.40% stop codons for TDD6, and 64.87 ± 1.66% stop codons for TDD14 (L4/L26 linker) respectively with similar results for three DddA split variants (Fig. 3C and Supplementary Data 10), and other TDDs including TDD21, TDD25 and TDD27 (Supplementary Fig. 3d). We also noticed increased editing efficiencies at positions C2 and C5 for TDD14 when tested as a ZF-CBE-nickases (Supplementary Fig. 6a). Taken together, these results introduce additional dsDNA deaminase domains with a variety of different substrate preferences and base editing profiles that can be fused to ZFs, TALEs or other DNA binding domains for efficient cytosine base editing in eukaryotic cells.

## Knocking out CIITA in human T cells

Given the encouraging results of our TDDs in an immortalized cell line (human K562 cells), we decided to compare the performance of TDD6 and TDD14 to DddA$_{tox}$ for knocking out the CIITA gene in primary human T cells (Fig. 4A and Supplementary Fig. 7a). We noted varying levels of cell viability across the tested deaminase domains where DddA$_{tox}$-G1333 and TDD6 showed the strongest negative impact on growth, and therefore suggested either a genome-wide specificity issue or overt toxicity, while cell growth in T cells transfected with DddA$_{tox}$-G1397, DddA$_{tox}$-G1404 and TDD14 was indistinguishable from that of untreated samples (Fig. 4B, Supplementary Fig. 7b and Supplementary Data 11). For constructs with good cell viability, we observed similar editing activity levels as in K562 cells using a PCR-based NGS assay, for example up to 80.81 ± 0.44% stop codons for DddA$_{tox}$-G1404, and up to 73.62 ± 0.18% stop codons for TDD14 (Fig. 4C, Supplementary Fig. 7c and Supplementary Data 12). The unintended editing events at positions C2 and C4 were present with comparable efficiencies as noted before in K562 cells (Supplementary Fig. 6b). CIITA is a transcriptional coactivator that regulates γ-interferon-activated transcription of Major Histocompatibility Complex (MHC) class I and II genes[37]. Thus, we confirmed that the stop codons introduced into the CIITA transcript diminished the percentage of cells expressing a MHC class II cell surface protein (Fig. 4D, Supplementary Fig. 7d and Supplementary Data 13). Taken together, these results suggest that ZF-CBEs can be used to efficiently knock out genes in human T cells. Our data also suggest that the genome-wide specificity profile of different dsDNA deaminases should be characterized further and potentially improved to be suitable for therapeutic applications.

## Genome-wide specificity profiling of ZF-CBEs

Encouraged by the performance of ZF-CBEs containing DddA$_{tox}$-G1404 and TDD14 at the CIITA locus in T cells, we profiled the genome-wide specificity of these constructs by adapting published techniques[38,39] to detect ZF-CBE off-target sites (Fig. 5A). This genome-wide activity assay utilizes purified genomic DNA and purified ZF-CBEs and then converts edited uridine bases into single-stranded DNA breaks that are detected by enrichment of specific DNA breakpoints using whole-genome sequencing. We identified 574 and 98 candidate sites for DddA$_{tox}$-G1404 and TDD14 respectively. The intended target at CIITA as well as 36 additional candidate sites were observed for both ZF-CBEs (Fig. 5B). This overlap was unsurprising as both ZF-CBEs utilize the same ZF arrays.

We next assayed cellular base editing activity in human T cells at these candidate off-target sites for each ZF-CBE using RNase H-dependent multiplexed PCR and sequencing (rhAmpSeq™)[40]. This method utilizes RNA-base-containing blocked primers that are unblocked by the RNase H2 enzyme only when bound to the intended template. This reduces primer dimer formation and increases multiplexed PCR specificity, making it an ideal method to profile large pools of off-target editing events. Primers targeting CIITA and all 634 candidate off-target sites for both DddA$_{tox}$-G1404 and TDD14 were used for primer panel generation. Given the partial overlap of sites identified with the genome-wide specificity assay between ZF-CBEs and the use of the same zinc fingers in each design, we tested all loci across both ZF-CBEs used to identify these sites as well as a related DddA$_{tox}$ construct DddA$_{tox}$-G1397. With rhAmpSeq™, we obtained adequate numbers of sequence reads to accurately measure cytosine base editing at 458 candidate off-target sites (Fig. 5C). In the presence of the nickase construct, we observed editing above 1% for 343 loci with DddA$_{tox}$-G1397, 314 loci with DddA$_{tox}$-G1404, and 153 loci with TDD14. In addition, the nickase construct only increased base editing signal at the intended on-target site (Supplementary Fig. 8a), implying such boosts to the base editing signal can be very specific with the ZF-CBE-nickase architecture used in this study. Lastly, both DddA ZF-CBE split constructs exhibited higher levels of editing compared to TDD14 at off-target sites (Supplementary Fig. 8b) with the highest being ~87% for DddA$_{tox}$-G1397, ~62% for DddA$_{tox}$-G1404, and ~16% for TDD14.

Considering the overlap in off-target editing activity between DddA$_{tox}$ split constructs and TDD14, we next explored how much of the off-target editing could be explained by ZF binding to DNA sequences near the observed off-target site. Within off-target events with >1% base editing levels for each ZF-CBE we searched for the most prominent sequence motifs across the entire amplicon. We identified a

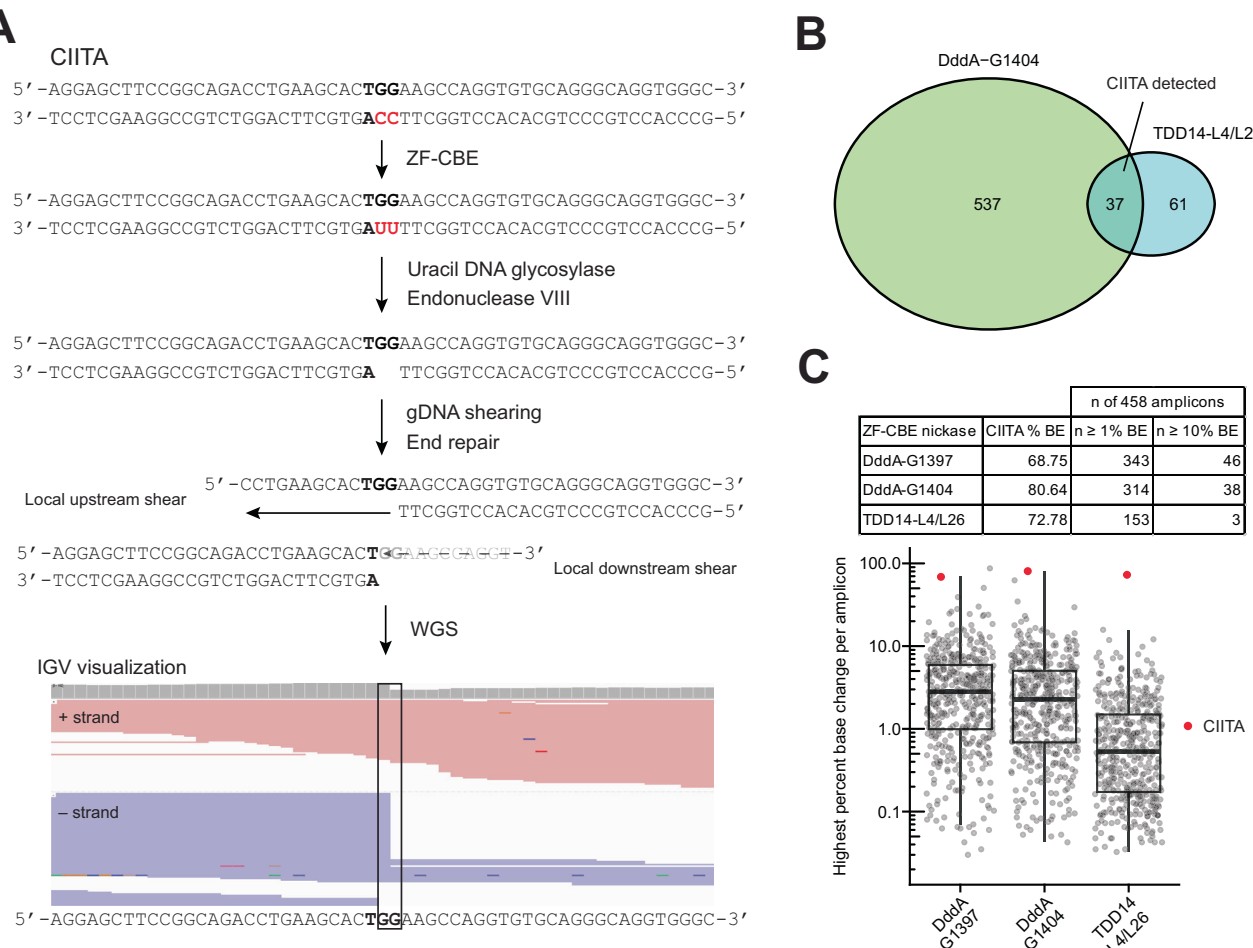

**Fig. 5 | Genome-wide specificity analysis of ZF-CBEs targeting the CIITA locus.**
**A** Overview of the genome-wide specificity assay. Purified genomic DNA and ZF-CBEs are co-incubated, converting on-target and off-target cytosines to uracils. USER, a combination of uracil DNA glycosylase and endonuclease VIII, then excises uridines, leaving ssDNA breaks at base edited sites. The genome is then sheared and end repaired, introducing random breaks throughout the genome along with common breaks at base edited sites. Whole-genome sequencing is then used to identify these common breaks (observed as identical 5′ read start positions with at least 10 reads and representing at least 20% of the total reads per position) corresponding to on and off-target base edits. CIITA base editing by TDD14-L4/L26 is indicated in the IGV visualization. **B** The genome-wide specificity assay identified 574 candidate off-target sites for DddA$_{tox}$-G1404, 98 for TDD14-L4/L26, and 37 of these were shared between ZF-CBEs. CIITA was observed in the candidate lists for both ZF-CBEs. **C** Base editing at off-target sites identified in the genome-wide specificity assay was measured in human T cells using rhAmpSeq™. DddA$_{tox}$-G1397, DddA$_{tox}$-G1404, and TDD14-L4/L26, all with nickase, were screened for base editing at 458 candidate off-target sites identified in all genome-wide specificity assays. The highest percent change of an A or T nucleotide per each amplicon is shown as an indicator of cytosine base editing on either strand. Box plot elements: Center line, median; box limits, upper and lower quartiles; whiskers, 1.5x interquartile range; all data points shown. Source data are provided as a Source Data file.

sequence motif with strong similarity to the ZF-CBE-R CIITA binding site in 97–100% of these sites for all ZF-CBEs (Supplementary Fig. 8c). We also identified a weaker motif matching a portion of the ZF-CBE-L binding site when using a narrower search window surrounding the highest edited base (Supplementary Fig. 8d). This implies many of the identified off-target events were caused by the ZF protein (ZFP) and repeating the rhAmpSeq panel with ZF-free CBEs indicated no editing above 1% (Supplementary Fig. 9). Thus, we expect decreasing ZFP binding affinity would reduce such off-target editing. Additionally, we identified a strong 5′-TC/C preference for all ZF-CBEs when centering the search on the highest edited base, although this preference was slightly stronger for DddA$_{tox}$ constructs than for TDD14 (Supplementary Fig. 8e). TDD14 also indicated a weaker 5′-AC preference along with weaker preference for the second C compared to DddA$_{tox}$ constructs, confirming what was observed at other genomic sites (Supplementary Fig. 4 and Supplementary Data 8). Taken together, while most of the off-target editing measured here was ZFP-dependent, the magnitude in variation of editing at these sites highlight that having a collection of TDD variants with different base editing properties is

critical for engineering ZF-CBEs with the required specificity for therapeutic applications.

## Discussion

In this study, we identify and characterize multiple toxin-derived cytidine deaminases capable of generating efficient CBEs without the need to unwind dsDNA. This allows these deaminase domains to be fused to any protein-guided DNA binding protein such as ZFs or TALEs for efficient editing of eukaryotic genomes. The most promising deaminase for therapeutic applications, TDD14, enabled up to ~73% editing of the CIITA gene in human T cells with low levels of indels (<0.2%), good cell viability and a favorable genome-wide specificity profile when delivered in conjunction with a ZF nickase. We confirmed a strong preference of DddA$_{tox}$ for 5′-TC target sites while other TDDs, including TDD6 and TDD14, showed an increased targeting density due to 5′-DC compatibility (D = A, G or T). Notably, we also observed that some TDDs including TDD14 had a strong editing preference for the first cytosine within a 5′-TCC substrate while DddA$_{tox}$ tends to edit both cytosines effectively. Our data also suggest that TDD14 has a

similar 5′-nt preference as TDD6 but with a narrower base editing window which likely reduces the number of potential off-target sites and therefore may explain why TDD6 and not TDD14 showed reduced cell viability in T cells. We also noticed substantially reduced cell viability for a DddA$_{tox}$-G1333-derived ZF-CBE in T cells as well as a higher number of off-target sites for the corresponding DddA$_{tox}$-G1397 and DddA$_{tox}$-G1404 variants when compared to TDD14. These results are in agreement with two recent studies where thousands of DNA off-target sites were observed in the nuclear genome of mouse embryos[41], and hundreds of DNA off-target events were detected in the nuclear genome of human cells[42] that were treated with DddA$_{tox}$-derived mitochondrial base editors. Notably, our ZF-CBEs require nickase activity at the tested CIITA site to show high on-target editing levels. This behavior implies that highly active off-target editing would also require off-target nicking in close proximity, especially when using a ZF-CBE that drives editing on only a single DNA strand. As off-target editing was not observed to increase with addition of the ZF-CBE-nickase construct, this implies high on-target editing can be achieved with low off-target editing levels when using a ZF-CBE with a narrower base editing window.

We envision that construct architecture improvement strategies that were previously described in the context of ZF nucleases (ZFNs)[43,44], and ZF-targeted transcriptional regulators[45], will guide future development of therapeutic ZF-CBEs. For example, high ZF design densities and different amino-terminal and carboxy-terminal linkers[43] will allow the optimal positioning of the deaminase domain with the intended target cytosine. Furthermore, the specificity profile of ZF-CBEs can be improved by using ZF variants with reduced non-specific DNA contacts as demonstrated for ZFNs with no detectable off-target editing in the human nuclear genome[44] and ZF-CBEs with improved specificity in the human mitochondrial genome[17]. We expect the former to have a large effect on off-target editing given the high sequence homology we observed between the ZF-CBE-R binding site and most of the off-target sites indicating base editing. Such improvements will also reduce off-target nicking introduced independently of base edits that could lead to indel formation. The level of off-target nicking was not measured in this study, but could be investigated for therapeutic applications by reversing the D450N FokI mutation and co-transfecting a dsDNA oligonucleotide to identify the resulting dsDNA breaks[44]. Lastly, while the ZF-CBE off-target editing observed here was ZF-dependent, independent CBE off-target editing has been observed for DddA using other genome-wide assays[42,46]. Notably, such effects have been reduced by introducing mutations to the interface of DddA$_{tox}$ halves to prevent spontaneous assembly[46]. We anticipate that similar strategies can be applied to the TDDs described in this study in the future if needed.

To show generalizability, we demonstrated that both our TDDs and our ZF-CBE-nickase architecture are compatible with other ZF design systems. We first replaced DddA with TDD6 and TDD14 in four previously established DddA-derived ZF-CBEs[23] and achieved comparable base editing performance at all target sites in K562 cells (Supplementary Figs. 12–15a, b and Supplementary Data 27–30). TDD14-derived ZF-CBEs targeted to all four sites did not have a major impact on cell viability when tested in T cells (Supplementary Fig. 16 and Supplementary Data 31). We then used publicly available ZF helices[47] to design additional ZFPs to convert selected DddA and TDD14-derived ZF-CBEs into ZF-CBE-nickases. Here, we were able to successfully establish ZF-CBE-nickases for all four target sites (Supplementary Figs. 12–15c, d). Taken together, this established the compatibility of our nickase architecture and TDD domains with alternative ZF design systems. We anticipate that recent advancements in ZF design through a universal deep-learning model[47] will further guide future ZF-CBE development efforts within the ZF research community.

Most notably, the compact construct architecture of ZF-CBE-nickases can make it possible to package all three components in a single AAV vector and thus shows promise for future therapeutic application in vivo. For example, the size of the TDD14-derived ZF-CBE-nickases described in this study can be reduced to ~3.65-kb by using a single UGI and removing unnecessary components such as FLAG-tags or redundant NLS sequences. Individual components could be separated by self-cleaving peptides and could therefore be controlled by the same regulatory elements[48]. The lack of an RNA guide also enables the delivery of ZFP-deaminase fusions by an exciting protein injection system[49] and presumably our ZF base editing architecture would also be compatible with this delivery method.

Potential applications for ZF-CBEs include the generation of knockout mutations, the correction of disease-causing mutations, and the induction of activating or deactivating mutations in regulatory elements. ZF-CBEs are better suited to simultaneously knock out multiple genes than nucleases because the intended edit is a precise knock out instead of a distribution of different indels. Multiple nuclease-induced double-strand breaks can also lead to translocations while our ZF-CBE data only shows background indel levels. Selection or enrichment strategies can be applied if a desired editing event is inefficient and the targeted cell line is susceptible to antibiotic treatment or FACS-mediated enrichment[50,51].

## Methods

### ZF-CBE constructs
DNA sequences for all ZF-CBE constructs can be found in Supplementary Data 15 and corresponding annotations can be found in Supplementary Data 16 (for TDDs) and Supplementary Data 17 (for other construct components). Most constructs were synthesized and cloned by Twist Biosciences utilizing their clonal genes service. The TDD variants of ZF-CBE-nickases were cloned using NEB's Gibson assembly protocol. Some constructs include base-skipping linkers between fingers within a ZFP array to increase ZF design options[43]. All FokI nickases are based on heterodimeric ELD and KKR FokI variants[35]. All constructs were sequence confirmed using Nextera XT DNA library prep kit (Illumina, FC-131-1096) to verify full plasmid sequences.

### Base editing in K562 cells
K562 cells (ATCC, CCL243) were cultured using RPMI-1640 growth medium supplemented with 10% FBS (Fetal Bovine Serum) and 1x PSG (Penicillin-Streptomycin-Gentamycin, Gibco, 10378-016) and maintained at 37 °C with 5% $CO_2$. ZF-CBEs and GFP were dosed as plasmid DNA (pDNA) in K562 cells. K562 cells were electroporated with pDNA using the SF cell line 96-well Nucleofector kit (Lonza, V4SC-2960) using manufacturer's protocol. Prior to electroporation, K562 cells were centrifuged at ~300 × $g$ for 5 min. Cells were resuspended at 2e5 cells per 12 μl of supplemented SF cell line 96-well Nucleofector solution. Twelve μl of cells were mixed with 8 μl of pDNA (400 ng of each ZF-CBE-L, ZF-CBE-R and optionally the nickase construct) and transferred to the Lonza Nucleocuvette plate. Nucleofector program 96-FF-120 was used to electroporate K562 cells with the pDNA mix on the Amaxa Nucleofector 96-well Shuttle System (Lonza). After electroporation, cells were incubated for 10 min at room temperature and transferred to a 96-well tissue culture plate containing 180 μl of complete medium (prewarmed to 37 °C). K562 cells were incubated for ~72 h and then harvested for base editing and indel quantification.

### PCR-based NGS assay for base editing and indel quantification
Seventy-two hours post transfection, cells were spun down at ~500 × $g$ for 5 min. Supernatant was discarded and 60 μl of QuickExtract DNA Extraction Solution (Lucigen) was added to each well. Genomic DNA was extracted by treating the cells to the following protocol using Accuprime HiFi reagents (Invitrogen), 65 °C for 15 min, 98 °C for 8 min. Target sites were amplified from the genomic DNA using the following PCR conditions, initial melt of 95 °C for 5 min; 30 cycles of 95 °C for 30 s, 55 °C for 30 s and 68 °C for 40 s; and a final extension at 68 °C for

10 min. Primers containing adapters (forward primer: ACAC-GACGCTCTTCCGATCT; reverse primer: GACGTGTGCTCTTCCGAT), targeting specific target sites were used at a final concentration of 0.1 μM. Sequences for the primers used can be found in Supplementary Data 18. The PCR productions obtained were then subjected to a second PCR to add Illumina barcodes to the PCR fragments generated in the first PCR. We used Phusion High-Fidelity PCR MasterMix with HF Buffer (NEB) for the second PCR and used the following PCR conditions, initial melt of 98 °C for 30 s; 12 cycles of 98 °C for 10 s, 60 °C for 30 s and 72 °C for 40 s; and a final extension at 72 °C for 10 min. PCR libraries generated from the second PCR were pooled and purified using QIAquick PCR purification kit (Qiagen). Samples were diluted to a final concentration of ~2 nM after they were quantified using the Qubit dsDNA HS Assay kit (Invitrogen). The libraries were then run on either an Illumina MiSeq using a standard 300-cycle kit or an Illumina NextSeq 500 using a mid-output 300-cycle kit using standard protocol. Unintended base editing events outside of the base editing window (defined as the gap between ZF array binding sites, including the first 5′ and 3′ flanking base) are listed in all Supplementary Data tables if they are 3% or higher. Similarly, unintended bases are listed if they are 2% or higher.

## mRNA production

AccuPrime Pfx DNA polymerase (Invitrogen) was used to generate the PCR template for mRNA production. The following conditions were used to run the PCR: initial melt of 95 °C for 3 min; 30 cycles of 95 °C for 30 s, 63.6 °C for 30 s and 68 °C for 3 min; and a final extension at 68 °C for 4 min. The primers were used at 0.4 μM final concentration and the sequences for the primers can be found in Supplementary Data 18. PCR template was purified using QIAquick PCR purification kit (Qiagen). mRNA was transcribed from the template using manufacturer's protocol from the 5x mMessage mMachine T7 ULTRA kit (Invitrogen). mRNA generated was purified using Agencourt RNA Cleanup XP beads using their standard protocol. mRNA was quantified using Quant-iT RNA Assay kit (Invitrogen).

## Base editing in human T cells

CD4+/CD8+ T cells were thawed and resuspended at 1e6 cells per milliliter in X-Vivo 15 Serum-free Hematopoietic Cell Media (Lonza, CAT#04-744Q) containing 5% Human AB Serum Heat-Inactivated (Valley Biomedical, #HP1022HI) and other supplements (detailed media composition is mentioned in Supplementary Data 19). Cells were activated for 72 h using Dynabeads human T-activator CD3/CD28 (cells to beads ratio of 1:3; Life Technologies, 11131D) and IL-2 (100 IU ml$^{-1}$; Thermo Fisher, CTP0023) without antibiotics. BTX H-200 transfection system (voltage 250 V, frequency 4 ms, # pulse 1 up to 12 depending on columns to be fed, set# of pulse at 1) was used to transfect mRNA for the ZF-base editors. 2e5 T cells per well were centrifuged at ~400 × g for 6 min. T cells were washed with 1x PBS and resuspended in BTXpress high performance electroporation solution (BTX, 45-0805) at 2e5 cells/100 ul/well I.e 2e6 cells/ml. In total, 100 μl of cells were mixed with 15 μl of mRNA (5 ug of each ZF-CBE-L, ZF-CBE-R and optionally the nickase construct) and transferred to the BTX 96-well disposable electroporation plate (2-mm gap, 125 μl, one plate, BTX, 45-0405-M). Electroporated cells were then transferred to a 48-well tissue culture plate containing 400 μl of complete medium and incubated at 30 °C and 5% CO$_2$ for 24 h. Cells were then moved to 37 °C with 5% CO$_2$ and incubator for another 6 days. Cells were then harvested for PCR-based base editing and indel analysis and flow cytometry.

## Flow cytometry

Approximately 2e5 T cells per well were harvested for flowcytometry. Cells were centrifuged at ~400 × g for 5 min and washed twice with 1x PBS containing 1% BSA at room temperature. Cells were incubated with 100 μl of eFluor 506 (Thermo, 65-0866-14) viability dye (diluted 1 to 1000 times) for 30 min at 4 °C in dark. After incubation, cells were washed twice with 1x PBS containing 1% BSA and incubated with 2.5 μl of PECy7 anti-human CD3 antibody (Biolegend, catalog number 300420, lot number B370636) and 2.5 μl of APC anti-HLA-DR antibody (Biolegend, catalog number 361714, lot number B289409) for 30 min at room temperature. After incubation, cells were washed twice with 1x PBS with 1% BSA and resuspended in 200 μl of 1x PBS with 1% BSA. Attune NxT acoustic focusing cytometer (Thermo Fisher) was used to perform flow cytometry and the FlowJo software was used to analyze the data generated. For gating strategy, refer to Supplementary Fig. 10.

## Search for toxin-derived deaminase domains

We used the sequence of the published DddA from *Burkholderia cenocepacia* as the search string in both BLAST and the metagenomics database at EMBL[52]. After filtering for redundant sequences, we chose to test sequences with a high degree of similarity to the active site residues of DddA. We noticed that some sequences with a lower degree of conservation contained a longer peptide sequence between the two active site cysteines, and these tended to form their own clade. We chose to test a small panel of deaminases from this separate clade TDDs 1-30 (Supplementary Data 16). To generate a consensus sequence, we used the consensus sequence finder webtool at kazlab.umn.edu[53] using TDD6 as the protein sequence search string.

## Protein purification of DddA$_{tox}$-G1404 and TDD14-L4/L26

To generate constructs for in vitro base editing activity assays, constructs encoding DddA$_{tox}$-G1404 were amplified by PCR from the mammalian expression vectors, without the UGI, and either a 6xHis or 10xHis tag was added using the amplification step, followed by assembly into an expression vector containing an MBP tag using the NEBuilder HiFi Assembly Master Mix (NEB #E2621X). Assembled constructs and plasmid sequences are indicated in Supplementary Data 15 with annotations in Supplementary Data 17. Constructs were then transformed into Nico21 cells (NEB # C2529H) following the suggested protocol. Transformed cells were grown overnight in 2xTY broth supplemented with 50 ug/ml kanamycin at 37 °C. One ml of overnight culture was then used to inoculate 50 ml 2xTY 50 ug/ml kanamycin with 100 μM ZnSO$_4$ and grown until the OD$_{600}$ reached 0.6−0.8. Cultures were then moved to an ice bath for 30 min before inducing with 0.5 ml 20% (w/v) L-Rhamnose. Protein expression was then carried out for 24 h at 16 °C before centrifugation and freezing of bacterial pellets at −20 °C. For protein purification, cell pellets were resuspended in 2.5 ml lysis buffer (20 mM HEPES pH 8.0 (Fisher Scientific # AAJ63578AP), 2 M NaCl, 5% glycerol, 1% CHAPS, 25 μM ZnSO$_4$, 1 mM TCEP (Fisher Scientific # AAH51864AC), and 1x Halt Protease and Phosphatase Inhibitor Cocktail, EDTA-free (Thermo Scientific #78441)). Bacteria were sonicated using a Q700 sonicator (Qsonica # Q700-110) with a ¼ inch probe with pulse 5 s on, 10 s off for a total time of 5 min, at 50% amplitude. Lysates were then centrifuged for 15 min at 22,700 × g and then diluted ten-fold with cold column loading buffer (5% glycerol, 20 mM HEPEES pH 8.0, 25 μM ZnSO$_4$, 1 mM TCEP). Lysate was then loaded onto a 1 ml HisTrap HP column (Cytiva) at 1 ml/min, washed with 20 ml running buffer (20 mM HEPES pH 8.0, 25 μM ZnSO$_4$, 1 mM TCEP, 500 mM NaCl), followed by 10 ml running buffer supplemented with 50 mM imidazole before eluting with running buffer supplemented with 150 mM imidazole. Constructs encoding TDD14-L4/L26 were similarly cloned and expressed, but ZF-CBE-R-ELD-D450N was purified via the maltose binding protein tag. Cleared lysate was added to a 5 ml MBPTrap HP column (Millipore-Sigma # GE28-9187-80) on the AKTA Pure at 1 ml/min. The column was washed with 12 column volumes of running buffer and eluted with 5 volumes of elution buffer (20 mM HEPES pH 8.0, 25 μM ZnSO$_4$, 1 mM TCEP, 500 mM NaCl, 50 mM Maltose). For all purified proteins, fractions were pooled and buffer

exchanged into storage buffer (20 mM 20 mM HEPES pH 8.0, 25 µM ZnSO$_4$, 1 mM TCEP, 500 mM NaCl, 50 mM L-glutamate, 50 mM L-arginine) by centrifugal filtration. Fractions were pooled and protein yield and purity was confirmed using a NuPage 4–12% Bis-Tris gel (Thermo Fisher #NP0321BOX) (Supplementary Fig. 11) alongside BSA dilutions. Protein was stored in 50% glycerol in −20 °C.

### ZF-CBE genome-wide specificity assay

GM24631 cells (Coriell) were grown and harvested according to the recommended conditions, and genomic extraction was performed using Qiagen Blood & Cell Culture DNA Maxi Kit (Qiagen #13362) following the recommended protocol. For all assays, GM24631 genomic DNA was added to a final concentration of 25 ng/µl in 10 mM Tris pH 7.5, 130 mM NaCl, 2 mM MgCl$_2$, 10 µM ZnSO$_4$, 1 mM DTT. For DddA$_{tox}$-G1404 CBE genome-wide specificity assay DddA$_{tox}$-G1404 -derived ZF-CBE-L and ZF-CBE-R-ELD-D450N were added to a final concentration of 17.4 nM and 177.1 nM, respectively, and incubated for 1 h at 37 °C. The TDD14 reaction was similarly set-up but TDD14-L4/L26-derived ZF-CBE-L and ZF-CBE-R-ELD-D450N were added to a final concentration of 40.7 nM and 95.8 nM, respectively. The unedited control was similarly set-up but without any ZF-CBE constructs. All treated samples then underwent an isopropanol/ethanol extraction. Sodium acetate was added to a final concentration of 0.3 M, glycogen (Thermo Fisher #R0561) was added to a final concentration of 1 µg/µl, then 0.65 volumes of isopropanol was added to the solution and mixed. The mixture was placed at −20 °C for 30 min then centrifuged at 12,000 × $g$ for 20 min at 4 °C. The supernatant was removed and 1 ml of 70% Ethanol at −20 °C was added. The mixture was centrifuged at 12,000 × $g$ for 10 min at 4 °C and the supernatant was again removed. EB (Qiagen #19086) was added and DNA was resuspended by light shaking at 37 °C overnight. The next day, samples were centrifuged under vacuum for 20 min with heat to remove excess ethanol. After ZFP-CBE incubation but before USER treatment, samples were checked for base editing activity via on-target amplification and next-generation sequencing. Briefly, 130 ng was used in the first PCR reaction and the protocol outlined in "PCR-based NGS assay for base editing and indel quantification" was followed. Base editing for these samples is reported in Supplementary Data 14. Next, samples underwent USER treatment. In total, 5.5 µg treated gDNA was incubated with 16.5 units USER® enzyme (NEB #M5505L) in a total volume of 275 µl for 3 h at 37 °C. 9.2 U of Proteinase K was added and samples were incubated for 5 min at 37 °C. Isopropanol/ethanol extraction, overnight resuspension, and removing excess ethanol was performed following the protocol above. Samples were prepped for WGS using the TruSeq DNA PCR-free prep kit (Illumina #20015962) following the protocol for 550 bp insert size on the Covaris M220 (Covaris #500295). Libraries were quantified using Qubit and the average amplicon size was estimated using sample run after genomic shearing with the Bioanalyzer High Sensitivity DNA assay (Agilent #5067-4626) on the Bioanalyzer 2100 expert and adding an additional 120 bp for adapter size. All samples were run on a NovaSeq6000 using a S2 300 cycle kit (Illumina #20028314).

### ZF-CBE genome-wide specificity assay analysis and identification of potential base edited sites

NGS reads were quality filtered (default $Q$-score 14) and aligned to hg38 with BWA MEM. Alignments were then filtered for mapping quality (MAPQ 50) and soft clipping of 5′ ends (≤5 bp allowed) to remove mapping artifacts as false signals. Optical duplicates were removed using Picard's MarkDuplicates tool. 5′ start sites were counted over a sliding window over the genome and kept as candidate off-target sites if passing minimum number of reads (default 10 reads) and cutoff threshold of total reads (default 20%). Candidates were then filtered out if positions overlapped repeat regions or shared with the unedited sample. Results are indicated in Supplementary Data 23–25.

Multiple candidates were grouped into a single candidate if within 50 bp of one another for further analysis and rhAmpSeq™ primer generation.

### rhAmpSeq™

Primer pools targeting potential base edited sites identified by the ZF-CBE genome-wide specificity assay were designed and synthesized by IDT using their rhAmpSeq™ Design Tool for CRISPR Gene Editing Analysis. Amplicon info is given in Supplementary Data 20–22. Potential base edited sites were amplified from genomic material isolated from transfected T cells using the rhAmpSeq™ CRISPR Library Kit (IDT #10007319). The recommended protocol was followed with some modifications. After transfection, genomic DNA was isolated using QuickExtract to a final concentration of 1440 cells/µl. In total, 6.4 µl of this solution was used with 1 µl DMSO in the first rhAmpSeq™ PCR reaction, adjusting the total reaction volume to 25 µl. In total, 10.2 µl of PCR 1 was used with 0.8 µl of DMSO in the second PCR reaction. Samples were sequenced to obtain 60,000-fold read coverage per amplicon on a NextSeq2000 using a P3 300 cycle kit (Illumina #20040561). Reads were processed and amplicons were demultiplexed using an in-house script using the first 23 bp of reads as detailed in ref. 44. Results are indicated in Supplementary Data 26 and 34 for data used in Fig. 5C and Supplementary Fig. 9, respectively. Amplicons were retained for analysis for each sample if present in ≥2 of 4 replicates with ≥1000 reads each. Amplicons were further removed if <15% of the reads resembled the expected hg38 sequence in the unedited sample. Positions with allele frequencies >0.04 and <0.96 in the unedited sample were also removed from base editing calculations.

### Design of zinc finger arrays using publically available information to add nickase capability to base editor constructs reported in Willis et al.[23]

Ichikawa et al.[47] recently published numerous examples of zinc finger constructs designed with their AI-guided tool (27 different zinc finger arrays with 8 zinc fingers per array and 37 different zinc finger arrays with 6 zinc fingers per array). Their system uses different zinc finger recognition helix sequences to target the same DNA triplet in different contexts, but in most cases the choice of helix for a given triplet appeared to depend mainly on the DNA base on the 3′ edge of the DNA triplet targeted by the helix. So we used the information they provided to generate a simple context-aware modular assembly scheme that considers the DNA triplet recognized by each zinc finger as well as the DNA base adjacent to the 3′ edge of the triplet. The information we used is shown in Supplementary Data 33. This system is simpler than their full ZFDesign algorithm, but still proved to be quite successful for designing zinc finger arrays to add nicking capability to four of the zinc finger base editor constructs presented by Willis et al. We expect that taking the base flanking the 3′ edge of the target triplet into account will perform better than standard zinc finger modular assembly because it accounts for more aspects of zinc finger context that is well established to be important for zinc finger target recognition[54,55].

### Motif analysis

Motifs were identified from the rhAmpSeq™ results using meme (version 5.5.0) from memesuite.

### Data visualization

GraphPad Prism 9, Adobe Illustrator 2022, Microsoft Excel 365 and RStudio were used for generating figures and tables.

### Reporting summary

Further information on research design is available in the Nature Portfolio Reporting Summary linked to this article.

## Data availability

NCBI accession numbers of deaminase used in this study are listed in Supplementary Data 16. Amino acid sequences and DNA sequences of ZF-CBEs used in this study are provided in Supplementary Data 15–17. Illumina sequencing data underlying all experiments have been deposited in the NCBI Sequence Read Archive under accession code PRJNA1052081. Source data are provided with this paper.

## Code availability

Custom computer code is available upon request although comparable analysis can be performed with publicly available software. This code was used for basic base editing and Indel analysis. Please contact the corresponding author, J.C.M., for access.

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

## Acknowledgements

We thank Lifeng Liu, Nga Nguyen, and Luis Rodriguez for assistance with next-generation sequencing. We also thank Sarah Hinkley, Stephen Lam, Emily Tait and Bryan Bourgeois for assistance with vector production and sequence validation, Ryann Swale for project management support, Jason Fontenot for helpful discussions and Sandy Macrae for encouragement and support.

## Author contributions

F.F., B.N.K., S.A.-F., J.E.D., A.R., P.L., G.D.D. and J.C.M. designed experiments. F.F., B.N.K., S.A.-F., J.E.D., V.V., N.J.S., C.N., Y.Z. and N.A.S. performed experiments. F.F., B.N.K., J.E.D., S.A.-F., A.R., P.L. and J.C.M. performed data analysis. F.F., J.E.D., A.R., P.L., G.D.D. and J.C.M. supervised experiments. J.E.D., G.L., J.E., Y.R.B., D.A.S. and J.C.M. developed custom computer code. S.A.-F. performed bioinformatics analysis to identify TDDs. F.F., B.N.K., J.E.D. and J.C.M. wrote the manuscript with input from all authors.

## Competing interests

All authors are full-time employees of Sangamo Therapeutics. Sangamo Therapeutics has filed patent applications regarding base editing systems described in this study, listing F.F., S.A.-F. and J.C.M. as inventors.
