## [Peer Review File · Nature Communications]

Reviewers' Comments:

Reviewer #1:

Remarks to the Author:

The authors of "A compact zinc finger architecture for highly efficient base editing in human cells" detail the development and design of a consistent and somewhat compact zinc finger - base editor. They show the generalizable improvement in on-target activity with the addition of a nick near the editing target and a clever architecture of arranged domains so that the entire tool is compatible with AAV delivery. They also explore new "Toxin Derived Deaminases" (TDD) to find new options with improved parameters through a homology driven metagenomic analysis. The authors clearly show improved activity over prior enzymes and approaches used for zinc finger or TALE base editors. However, the benefit of a smaller base editor is not entirely clear and the generalizability of the approach remains limited to the small set of groups with access to routine zinc finger design or production. While I think the advance offered in this manuscript will be appreciated by the community, there are a handful of issues that need to be addressed.

1. Impact and generalizability. I love the approach the authors took to find new base editors but their focus has been totally on using these with zinc fingers to accommodate compact delivery (see #2). But zinc fingers that bind novel targets have been notoriously difficult to generate beyond a very small group of specialized labs and companies. So the impact for the community could be quite limited unless there is a general resource for them to design their zinc fingers. Otherwise, the small size doesn't benefit the community that will still be reliant on the easy programmability of CRISPR and TALE reagents. Some examples that could be used, ZiFiT that came out of Keith Joung's lab, the more recent ZFDesign, or ZFs available through Sigma (the Sangamo designs), these are available to the community and would provide nice examples of how the community could take advantage of this work. But to my knowledge the authors have only demonstrated that their tools work with their own zinc fingers. Why is this a concern? Is there something critical about these zinc fingers? Do they have higher affinity? Better specificity? Etc from what the community has access to? If so, the impact of the tool is really limited to Sangamo, not the field. And, since using zinc fingers and TALEs for base editing is already established, the tool itself isn't novel. Therefore, I would like to see how their approach works with programmable zinc fingers that are available to the community. This will indicate just how impactful the advance might be. Alternatively, or in addition to, they might demonstrate that their new TDDs work with CRISPR tools that the community does have access to.

2. Small Cargo. The authors point out multiple times the advantage of the small size of this base-editor/nickase for AAV delivery. However, they never really suggest why this is an advantage. They are focused on T-cell modifications that could be useful therapeutically. But the modifications that they are suggesting are typically done ex-vivo and DNA can be delivered to T-cells by electroporation, LNPs, and with other forms of viruses. So noting the size of AAV's only really makes sense as an advantage if they are talking about in vivo delivery. If that is long-term suggestion here it is not clear in the manuscript. In addition, delivering a genome editing tool in vivo by AAV could have its own challenges including long-term expression of the cargo that could lead to off-target activity. Ideally one would have some sort of a kill switch to turn off the tool in this context but that would take more of the cargo in the AAV. One more point on the size advantage, in the discussion the authors note a protein delivery tool but Kreitz et al 2023. But in that paper they delivery SpCas9 with the same mechanism and show the same activity as zinc fingers. So size isn't really an advantage for that delivery. Regardless, the authors seem very fixed on size and AAV delivery yet they don't demonstrate a size advantage with any experiments so clearly not important for what they've actually done. But I do appreciate the point about size, the authors just need to make it clear in the text how size is an advantage and in what kind of applications would they really be using the size advantage for with AAV delivery that can't be done another way?

a. A table comparing the size of the zinc finger base editing/nicking system versus various new forms of Cas9 including the OMEGA constructs recently published by Feng Zhang's lab would be very helpful.

3. Adding a nickase with the dimeric fok1 domain could lead to off-target nuclease activity for the heterodimer or through homodimerization. The authors never really look for this so I questioned what form of fok1 they are using. After looking through the method I see they are using the EDL and KKR heterodimeric versions of the nuclease. This is exactly what I was hoping for! But it

should be clear in the text to make sure others that might want to try this approach will understand this is important to avoid off-target activity.

4. Speaking of the nickase, it seems the authors have looked for off-target activity based on potential deaminase events, but nicking the DNA can also lead to genomic instability and off-target lesions. Have the authors looked for these and what would they expect if the nickase is expressed for weeks or months with in vivo AAV delivery?

5. The authors note, "Our data suggests that linker optimization studies and testing both split orientations ... can both be effective options for fine-tuning the base editing profile within the base editing window." I would argue that this makes the programmability of the zinc fingers even more important. No one wants to have to optimize linkers and orientations for every possible target and therefore knowing an exact linker/orientation/spacing architecture that produces consistent activity would be far more useful. But this will also require the ability to generate zinc finger arrays for any target necessary to accommodate that architecture. This again motivates a request to demonstrate that this tool works with zinc fingers that will be available to the community.

Reviewer #2:

Remarks to the Author:

This in-depth study by Fauser F. et al describes an alternative approach to cytidine deamination with the intent of seamless gene disruption by directly targeting dsDNA as opposed to redirecting RNA editors to DNA using CRISPR machinery. This involves the use of toxin derived deaminases (TDD), recently reported for the editing of nuclear and mitochondrial DNA when combined with CRISPR, TALE or ZF motifs. The work presented here builds on this data with a primary focus on ZF fusion optimisations using existing and novel TDDs for improved on-target editing efficiency, control of window of activity and reduced genome wide off-target activity. Proof of concept has been demonstrated in primary human T cells where effective knockout of MHC Class II gene CIITA was achieved. This powerful platform offers broad clinical potential for targeted modification of the genome with increased specificity while one of the key advantages lies in its compact and modular design which can be incorporated and delivered by a single AAV vector.

Limitations

The conducted study presents a rigorous validation of ZF-CBEs fusing novel TDD domains for targeted gene knockout. The main limitation that arises surrounds the complexity of designing and validating three ZF binding domains for each genomic site of interest, selecting the optimal TDD for editing of the intended base given the sequence context preferences, as well as controlling the affinity of the ZF binding to balance on/Off-target editing. Knowledge obtained from the CIITA targeting will help generate a set of parameters that could be followed for a separate genomic site, however, it is unlikely that efficiency, fidelity will necessarily be comparable.

Strategy may not be ideal for site specific correction of bases considering the wide window of activity, context dependency. Perhaps optimal in known hot-spot mutations where validated ZF-CBE combinations can be designed without the requirement for modulation of parameters.

Minor comments

It would have been interesting to see the packaging and delivery of the three components within a single AAV vector as alluded to in the manuscript.

What was the rationale for selecting CIITA as the single genomic target chosen? While probably critical for allogeneic CAR T cell manufacture, additional targets such as TCRab or B2M would most likely still be essential.

Following on the point above, it would have been interesting to have demonstrated multiplexed edits. This would have allowed for measure of translocations considering there are still low level indels generated in the edited samples.

Have you measured the persistence of ZF-CBE protein expression post electroporation in T cells by western blotting similar to fig s11?

Could be beneficial to reader to briefly explain the advantage of including a longer base skipping linker in the ZF-CBEs in certain circumstances but not others.

It is unclear what the full window of base editing activity is, would that be the entire 21bp sequence between left and right ZFs? Consider adding a sentence to define this as conventional CBE, ABE, AID editors typically have a defined 4-8bp activity distal to the nick site.

What is the reason for the large variability between untreated and DddA-G1397, DddA-G1404 and TDD14-L4/L26 in Figure 4B? The cell viability drop is noticeable in TDD6 and DddA-G1333 but hard to interpret considering the variability in untreated and increase in cell density in the other groups.

Supplementary Figure 3D. Avoid using error bars with s.d. when presenting data from 2 biological replicates.

Has the off-target profile of the DddAtox or TDD14 in the absence of a ZF domain been investigated?

Reviewer #3:

Remarks to the Author:

In the manuscript entitled "A compact zinc finger architecture for highly efficient base editing in human cells", the authors made two major findings, the first is that a FokI-based nickase design improves base editing at one strand, the second is that they identified many novel TDDs with different sequence specificity, and showed one of them, TDD14, enabling high on-target specificity and low off-target activities, potentially due to a narrower editing window. While interesting and potentially important for therapeutics development, the major concern for the manuscript is that most of the conclusions are based on results for one gene. A few more genes should be tested, and preferably some of mitochondrial genes should also be tested to show that the design works generally in both nucleus and mitochondrial. Furthermore, the speculation about cytosine base conversions on the opposite strand increase base editing at the other strand needs experimental evidence, for example, through transgene constructs with mutations at potential base editing sites of opposite strand. Other minor concerns:

1. The sentence "this new base editor architecture is also compact enough to enable packaging in a single AAV vector" should be removed from the abstract, since the authors has not shown any data related to AAV.
2. The conclusions about linker should be more specific by pointing out what linkers or a range of linkers should be tested in future studies.
3. In terms of genome wide off-target effects, a comparison for ZF-CBE and ZF-CBE-nickase is needed to provide guidance for future applications with ZF-CBE-nickase.
4. The title of the manuscript is too general. Nickase and/or newly identified TDD should be included.

Reviewer #1 (Remarks to the Author):

The authors of “A compact zinc finger architecture for highly efficient base editing in human cells” detail the development and design of a consistent and somewhat compact zinc finger - base editor. They show the generalizable improvement in on-target activity with the addition of a nick near the editing target and a clever architecture of arranged domains so that the entire tool is compatible with AAV delivery. They also explore new “Toxin Derived Deaminases” (TDD) to find new options with improved parameters through a homology driven metagenomic analysis. The authors clearly show improved activity over prior enzymes and approaches used for zinc finger or TALE base editors. However, the benefit of a smaller base editor is not entirely clear and the generalizability of the approach remains limited to the small set of groups with access to routine zinc finger design or production. While I think the advance offered in this manuscript will be appreciated by the community, there are a handful of issues that need to be addressed.

We thank the reviewer for their practical assessment of our work and for suggestions for how we can better demonstrate the accessibility of our work for the entire scientific community. Our point-by-point response is below:

1. Impact and generalizability. I love the approach the authors took to find new base editors but their focus has been totally on using these with zinc fingers to accommodate compact delivery (see #2). But zinc fingers that bind novel targets have been notoriously difficult to generate beyond a very small group of specialized labs and companies. So the impact for the community could be quite limited unless there is a general resource for them to design their zinc fingers. Otherwise, the small size doesn't benefit the community that will still be reliant on the easy programmability of CRISPR and TALE reagents. Some examples that could be used, ZiFiT that came out of Keith Joung's lab, the more recent ZFDesign, or ZFs available through Sigma (the Sangamo designs), these are available to the community and would provide nice examples of how the community could take advantage of this work. But to my knowledge the authors have only demonstrated that their tools work with their own zinc fingers. Why is this a concern? Is there something critical about these zinc fingers? Do they have higher affinity? Better specificity? Etc from what the community has access to? If so, the impact of the tool is really limited to Sangamo, not the field. And, since using zinc fingers and TALEs for base editing is already established, the tool itself isn't novel. Therefore, I would like to see how their approach works with programmable zinc fingers that are available to the community. This will indicate just how impactful the advance might be. Alternatively, or in addition to, they might demonstrate that their new TDDs work will with CRISPR tools that the community does have access to.

We agree with the reviewer that designing ZFNs using publicly available information has historically been very challenging. However, the zinc finger design requirements for split deaminase fusions are more flexible than the requirements for early ZFNs that required a precise orientation and spacing between the binding sites of the two zinc finger arrays fused to the FokI cleavage domains. This added flexibility for zinc finger-split deaminase fusions allowed Willis *et al.* to successfully edit multiple locations within the human mitochondrial and nuclease genomes using zinc fingers designed with a slightly modified version of the standard zinc finger modular assembly method (Willis *et al.*, 2022; Nat. Comm.) As an initial test of whether our novel deaminase domains would be compatible with zinc fingers accessible to the broader scientific community, we fused two of our best characterized deaminase domains, TDD6 and

TDD14, to zinc finger arrays described by Willis *et al.* that target five different genes in the human nuclear genome (EMILIN1, TRAM1L1, HBB, COL5A1, and DCAF8L2). We observed detectable activity at all five of these sites in human K562 cells using both TDD6 and TDD14, but it was challenging to PCR amplify the DCAF8L2 cleanly enough to accurately quantify base editing at this locus so we focused our efforts on the other four sites (EMILIN1, TRAM1L1, HBB, COL5A1). The data for base editing at these four sites is summarized in supplementary figures 12-16, and supplementary tables 27-32. Thus, we are able to clearly demonstrate that our novel deaminase domains could be fused to zinc fingers designed using publically available methods to create base editing reagents active in human K562 cells. But we acknowledge the possibility that zinc finger arrays designed by publically available methods might not be specific enough for use in more challenging cell types such as primary human T cells. To test if our deaminase domains fused to publically available zinc fingers would be specific enough for applications in primary human T cells, we utilized the same cell growth assay we used in Figure 4b to compare our zinc finger arrays to zinc finger arrays reporting by Willis *et al.* when both were fused to our best characterized deaminase domain TDD14. In general, the zinc finger arrays from Willis *et al.* fused to TDD14 are well tolerated in T cells. EMLIN2 and HBB do show a modest effect on growth rate, but this effect is much smaller than the effect of using a different deaminase domain such as TDD6 or DddA split at residue 1333 and shouldn't be extreme enough to prevent their use by the scientific community for research applications in useful cell types. And in the event that this modest decrease in growth rate isn't sufficient for a desired application, the specificity profile of ZF-CBEs can be further improved by using ZF variants with reduced non-specific DNA contacts. This was originally demonstrated for ZFNs targeting the human nuclear genome by our group (Miller *et al.*, 2019), and later adapted to ZF-CBEs with improved specificity in the human mitochondrial genome (Lim *et al.*, 2022). Thus, we conclude that both TDD6 and TDD14 will be useful for research applications in immortalized cell lines when fused to publically available zinc fingers and that TDD14 will be useful for applications with more stringent specificity requirements such as editing primary human T cells when fused to zinc finger domains accessible to the entire scientific community.

In order to add nicking capability to the ZF base editors targeted to EMILIN1, TRAM1L1, HBB, and COL5A1 we utilized publically available zinc finger design information reported after Willis *et al.* was published (Ichikawa *et al.*, 2023; Nat. Biotech.) to create a simple context-dependent zinc finger modular assembly scheme that anyone in the scientific community can easily replicate. We successfully used this to create active nickases that boosted the activity of the base editing reagents for all four of the target sites from Willis *et al.* that we investigated. The data is summarized in supplementary figures 12-15, and supplementary tables 27-30. Through both our novel deaminases and adding nicking capabilities we were able to outperform the original DddA reagents reported by Willis *et al.* at all four sites we investigated. Since this demonstration used only information that will be publically available once our manuscript is published, we feel confident that others will be able to utilize our methods to successfully edit genes they are interested in. To make it as easy as possible for others to replicate the process we used to add nicking capability to these base editors, the information from Ichikawa *et al.* we used in our design efforts has been placed in supplementary table 33.

Please note that we did not have sufficient time to negotiate a materials transfer agreement to obtain access to the full ZF Design tool for generating new data to revise our manuscript- presumably the full ZF Design tool would also have been capable of designing zinc finger arrays to add nickase functionality to ZF base editors.

2. Small Cargo. The authors point out multiple times the advantage of the small size of this base-editor/nickase for AAV delivery. However, they never really suggest why this is an advantage. They are focused on T-cell modifications that could be useful therapeutically. But the modifications that they are suggesting are typically done ex-vivo and DNA can be delivered to T-cells by electroporation, LNPs, and with other forms of viruses. So noting the size of AAV's only really makes sense as an advantage if they are talking about in vivo delivery. If that is long-term suggestion here it is not clear in the manuscript. In addition, delivering a genome editing tool in vivo by AAV could have its own challenges including long-term expression of the cargo that could lead to off-target activity. Ideally one would have some sort of a kill switch to turn off the tool in this context but that would take more of the cargo in the AAV. One more point on the size advantage, in the discussion the authors note a protein delivery tool but Kreitz et al 2023. But in that paper they delivery SpCas9 with the same mechanism and show the same activity as zinc fingers. So size isn't really an advantage for that delivery. Regardless, the authors seem very fixed on size and AAV delivery yet they don't demonstrate a size advantage with any experiments so clearly not important for what they've actually done. But I do appreciate the point about size, the authors just need to make it clear in the text how size is an advantage and in what kind of applications would they really be using the size advantage for with AAV delivery that can't be done another way? A table comparing the size of the zinc finger base editing/nicking system versus various new forms of Cas9 including the OMEGA constructs recently published by Feng Zhang's lab would be very helpful.

We agree with Reviewer 1 that our initial submission overemphasized the size advantage, especially in the context of the performed experiments. Consequently, we removed the sentence about AAV packaging from the abstract. We also added minor changes to the single construct AAV delivery sentences in the introduction and discussion sections to make it clearer that single construct AAV packaging of base editors with nicking capability is a potential future development step for in vivo applications.

We are commenting on long-term expression effects below in the context of Reviewer 1's 4th comment.

Regarding Kreitz et al: A therapeutically relevant delivery platform is ideally capable of delivering all necessary components to the target cell. While it is true that this new delivery platform can deliver SpCas9, it still requires the guide RNA to be stably expressed in the host cell. In contrary, the authors were able to deliver a fully functional Zinc Finger Base Editor to the host cell. We changed "compact size and lack of an RNA guide" to "lack of an RNA guide" in the discussion to address Reviewer 1's concern, and to avoid any size-related confusions on this matter for the reader.

3. Adding a nickase with the dimeric fok1 domain could lead to off-target nuclease activity for the heterodimer or through homodimerization. The authors never really look for this so I questioned what form of fok1 they are using. After looking through the method I see they are using the EDL and KKR heterodimeric versions of the nuclease. This is exactly what I was hoping for! But it should be clear in the text to make sure others that might want to try this approach will understand this is important to avoid off-target activity.

We would like to thank Reviewer 1 for pointing this out. We now emphasize the use of heterodimeric FokI variants in the results section, in addition to the Material and Methods section: "All FokI nickases

described in this study are based on heterodimeric ELD and KKR FokI variants (Doyon *et al.*, 2011) to avoid the assembly of an active nuclease through homodimerization at potential off-target sites.”

4. Speaking of the nickase, it seems the authors have looked for off-target activity based on potential deaminase events, but nicking the DNA can also lead to genomic instability and off-target lesions. Have the authors looked for these and what would they expect if the nickase is expressed for weeks or months with in vivo AAV delivery?

We appreciate Reviewer 1’s concerns about nickase-induced off-targets. We added the following sentence to the discussion section: “Such improvements will also reduce off-target nicking introduced independently of base edits that could lead to indel formation. The level of off-target nicking was not measured in this study, but could be investigated for therapeutic applications by reversing the D450N FokI mutation and co-transfecting a dsDNA oligonucleotide to identify the resulting dsDNA breaks (Miller *et al.*, 2019).”

We would expect therapeutic reagents to be highly specific with no to minimal impact on genome stability – even when expressed for weeks or months. If not possible, a kill switch could be installed in the expression cassette where e.g. a stop codon is induced in the ZF-CBE-Nickase open reading frame. We did not discuss kill switches since this is a challenge that applies to all genome editing reagents that are delivered using AAV, and not just ZF-CBEs. We envision that the reagents described in this study (TDDs & CBE-Nickase architecture) will predominantly be used outside of therapeutic applications where such concerns are of much less significance. For applications where high levels of specificity are required, we point the reader to specificity-improving studies (Miller *et al.*, 2019) that already showed to have beneficial effects when applied to ZF-CBEs (Lim *et al.*, 2022).

5. The authors note, “Our data suggests that linker optimization studies and testing both split orientations ... can both be effective options for fine-tuning the base editing profile within the base editing window.” I would argue that this makes the programmability of the zinc fingers even more important. No one wants to have to optimize linkers and orientations for every possible target and therefore knowing an exact linker/orientation/spacing architecture that produces consistent activity would be far more useful. But this will also require the ability to generate zinc finger arrays for any target necessary to accommodate that architecture. This again motivates a request to demonstrate that this tool works with zinc fingers that will be available to the community.

We agree with Reviewer 1 that high Zinc Finger design densities are important. We would like to refer to our response to Reviewer 1’s first comment regarding generalizability and the use of publicly available Zinc Finger design tools.

Reviewer #2 (Remarks to the Author):

This in-depth study by Fauser F. et al describes an alternative approach to cytidine deamination with the intent of seamless gene disruption by directly targeting dsDNA as opposed to redirecting RNA editors to DNA using CRISPR machinery. This involves the use of toxin derived deaminases (TDD), recently reported for the editing of nuclear and mitochondrial DNA when combined with CRISPR, TALE or ZF motifs. The work presented here builds on this data with a primary focus on ZF fusion optimisations using existing and novel TDDs for improved on-target editing efficiency, control of window of activity and reduced genome wide off-target activity. Proof of concept has been demonstrated in primary human T cells where effective knockout of MHC Class II gene CIITA was achieved. This powerful platform offers broad clinical potential for targeted modification of the genome with increased specificity while one of the key advantages lies in its compact and modular design which can be incorporated and delivered by a single AAV vector.

We are pleased to hear that Reviewer 2 recognizes our ZF-CBE-Nickase architecture as a powerful platform with broad clinical potential, and that the manuscript was well-received. We would like to thank reviewer 2 for their constructive feedback. We address the reviewer's comments below:

Limitations

The conducted study presents a rigorous validation of ZF-CBEs fusing novel TDD domains for targeted gene knockout. The main limitation that arises surrounds the complexity of designing and validating three ZF binding domains for each genomic site of interest, selecting the optimal TDD for editing of the intended base given the sequence context preferences, as well as controlling the affinity of the ZF binding to balance on/Off-target editing. Knowledge obtained from the CIITA targeting will help generate a set of parameters that could be followed for a separate genomic site, however, it is unlikely that efficiency, fidelity will necessarily be comparable.

We recognize Reviewer 2's concern about the complexity of designing and validating three ZF binding domains for each genomic site of interest. We'd like to point Reviewer 2 to our response to Reviewer 1's first comment on generalizability. Using zinc finger helices published by Ichikawa *et al.* (2023) we were able to successfully add nickases to zinc finger base editors targeted to four different human genes presented in Willis *et al.* (2022). We believe that the scientific community will have a high success rate designing such reagents when making full use of publicly available Zinc Finger design platforms.

We agree that the knowledge obtained from the reagents described in this study, as well as studies published by other labs, will help generate a set of parameters that will further streamline the design process of ZF-CBEs and ZF-CBE-Nickases. Our revised manuscript contains many additional examples of base editors utilizing our novel deaminase domains both with and without nickases with results ranging from highly active without a nickase (EMLIN2) to requiring a nickase for good activity (TRAM1L1 and HBB) to only obtaining decent activity with DddA and TDD6 (COL5A1) that will inform choices of deaminase domain and target site for future studies.

Strategy may not be ideal for site specific correction of bases considering the wide window of activity,

context dependency. Perhaps optimal in known hot-spot mutations where validated ZF-CBE combinations can be designed without the requirement for modulation of parameters.

We agree with Reviewer 2 that ZF-CBEs only have restricted use for reversing e.g. disease-causing mutations. We see the main advantage of ZF-CBEs in the induction of stop codons for knock-out applications, either *ex vivo* or *in vivo*. For the majority of knock-out targets, we anticipate that there is sufficient design space for the scientific community to design Zinc Finger-derived CBE-Nickases.

Minor comments

It would have been interesting to see the packaging and delivery of the three components within a single AAV vector as alluded to in the manuscript.

We agree with Reviewer 2 that experimental evidence for AAV packaging would have been interesting, but it was out of scope for this study.

We removed the single construct AAV delivery note from the abstract to put less emphasis on something without supporting data. We also added minor changes to the single construct AAV delivery sentences in the introduction and discussion sections to make it clearer that single construct AAV packaging is a potential future development step.

What was the rationale for selecting CIITA as the single genomic target chosen? While probably critical for allogeneic CAR T cell manufacture, additional targets such as TCRab or B2M would most likely still be essential.

We consider this manuscript as a platform development study with a strong emphasis on new TDDs and the novel ZF-CBE-Nickase construct architecture. Hence, we decided to target CCR5 since this site has been targeted with TALE-DddA before, and CIITA as an independent additional site.

Following on the point above, it would have been interesting to have demonstrated multiplexed edits. This would have allowed for measure of translocations considering there are still low level indels generated in the edited samples.

We agree with Reviewer 2 that this would be interesting in the context of a more therapeutic application-focused study, and that an in-depth evaluation of translocations between different base editor target sites is of utmost importance during the development of clinical-grade reagents.

Have you measured the persistence of ZF-CBE protein expression post electroporation in T cells by western blotting similar to fig s11?

No, we have not.

Could be beneficial to reader to briefly explain the advantage of including a longer base skipping linker in the ZF-CBEs in certain circumstances but not others.

We would like to thank Reviewer 2 for pointing this out. We added a sentence in the Material & Methods section to explain the advantage of base skipping linkers: "Some constructs include base-skipping linkers

between fingers within a ZFP array to increase ZF design options (Paschon *et al.*, 2019).”

It is unclear what the full window of base editing activity is, would that be the entire 21bp sequence between left and right ZFs? Consider adding a sentence to define this as conventional CBE, ABE, AID editors typically have a defined 4-8bp activity distal to the nick site.

We acknowledge that it is much harder to define the base editing window for ZF-CBEs or TALE-CBEs compared to CRISPR-derived base editors. We define the base editing window as “the gap between ZF array binding sites, including the first 5’ and 3’ flanking base” in the material and method section. However, the effective base editing window depends on the TDD (see Figure 3), and other parameters such as the linkers or the sequence composition within the base editing window. Target site specific nomenclatures are introduced in the corresponding supplementary figures, e.g. supplementary figure 2 for our main CIITA target site.

What is the reason for the large variability between untreated and DddA-G1397, DddA-G1404 and TDD14-L4/L26 in Figure 4B? The cell viability drop is noticeable in TDD6 and DddA-G1333 but hard to interpret considering the variability in untreated and increase in cell density in the other groups.

We agree that the cell density graph is hard to interpret, and we appreciate Reviewer 2 asking for clarification on this matter. We believe that this variability was introduced during sample handling steps for NGS and flow cytometry analysis between day 7 and day 10. However, it confirmed results from a pilot experiment, and results that we have obtained during the revision process (supplementary figure 16, supplementary tables 31-32).

Most importantly, the difference in cell density between various ZF-CBE reagents initiated our work on a genome-wide specificity assay to determine the off-target profile of these reagents.

Supplementary Figure 3D. Avoid using error bars with s.d. when presenting data from 2 biological replicates.

We removed the error bars from SI Figure 3D as suggested by Reviewer 2.

Has the off-target profile of the DddAtox or TDD14 in the absence of a ZF domain been investigated?

We appreciate Reviewer 2 for pointing out this concern, especially in light of relevant work exploring TALE-free DddA_{tox} off-target editing (Lee *et al.*, 2023, Lei *et al.*, 2022). We explored if any of the off-target editing reported with our ZF-CBEs in this study were ZF-independent editing events by repeating the rhAmpSeq on K562 cells transfected with ZF-free CBE constructs as seen in the new **Supplementary Figure 9a**. For both DddA_{tox}-G1404 and TDD14, we did not observe editing above 1% at any of the identified off-targets. We acknowledge that there may still indeed be ZF-free off-targets that were not identified in the genome-wide specificity assay that we do not analyze here. If this were the case, we anticipate explorations of similar mutations to split CBE halves that reduce spontaneous assembly, as performed in Lee *et al.*, 2023, could reduce such effects in translating this work to therapeutic applications.

Reviewer #3 (Remarks to the Author):

In the manuscript entitled "A compact zinc finger architecture for highly efficient base editing in human cells", the authors made two major findings, the first is that a FokI-based nickase design improves base editing at one strand, the second is that they identified many novel TDDs with different sequence specificity, and showed one of them, TDD14, enabling high on-target specificity and low off-target activities, potentially due to a narrower editing window.

We are delighted to hear that Reviewer 3 recognizes the importance of our new TDDs, and the FokI based nickase design. We would like to thank Reviewer 3 for their thoughtful comments. We address the reviewer's comments below:

While interesting and potentially important for therapeutics development, the major concern for the manuscript is that most of the conclusions are based on results for one gene. A few more genes should be tested, and preferably some of mitochondrial genes should also be tested to show that the design works generally in both nucleus and mitochondrial.

We recognize Reviewer 3's concern and included additional supporting data in the revised manuscript. We would like to point Reviewer 3 to our response to Reviewer 1's first comment. Our revised manuscript now includes editing at a total of 8 different sites in 6 different human genes (CCR5, EMILIN1, TRAM1L1, HBB, COL5a, and three locations in CIITA). We believe that this provides sufficient evidence that ZF-CBEs can be robustly targeted to desired target sequences.

Targeting mitochondrial genes was out of scope of this study. We are confident that the scientific community will be able to successfully transfer our TDDs to mitochondrial targets.

Furthermore, the speculation about cytosine base conversions on the opposite strand increase base editing at the other strand needs experimental evidence, for example, through transgene constructs with mutations at potential base editing sites of opposite strand.

We appreciate Reviewer 3 for pointing out the need for more evidence for this statement. Upon review, we realized the theoretical 50% base editing maximum stated in the paper has not been established as a standard rule for dsDNA cytosine base editing and other works have indicated > 50% editing without nicking of the strand opposite of the base edit (Boyne *et al.*, 2022, Willis *et al.*, 2022). We anticipate further mechanistic approaches will help elucidate how such high cytosine base editing occurs and can be used advantageously for therapeutic or research applications. As the high editing observed here does not seem to be an anomaly and we did not want to propose mechanistic insights without substantial evidence, we have removed the text and associated supplemental figure that discusses base edits on the opposite strand.

Other minor concerns:

1. The sentence "this new base editor architecture is also compact enough to enable packaging in a single AAV vector" should be removed from the abstract, since the authors has not shown any data related to AAV.

We would like to thank Reviewer 3 for bringing this to our attention, and we removed this sentence from the abstract as suggested.

2. The conclusions about linker should be more specific by pointing out what linkers or a range of linkers should be tested in future studies.

We appreciate Reviewer 3's comment on providing the reader with guidance for linker usage. We don't think that we have sufficient data to provide a solid recommendation for linker length. However, we have observed a high success rate when designing reagents based on our L26 linker, and we added a sentence to the result section to reflect that: "We recommend starting ZF-CBE designs with the L26 linker and exploring the effect of shorter linkers if necessary."

3. In terms of genome wide off-target effects, a comparison for ZF-CBE and ZF-CBE-nickase is needed to provide guidance for future applications with ZF-CBE-nickase.

Regarding a comparison between ZF-CBE and ZF-CBE-nickase on editing in the identified off-targets, we have explored this effect in the paper in **supplemental figure 8a** where we observe that addition of the nickase construct to ZF-CBEs only increased on-target editing. We realized that the original reference to this figure in the text and corresponding figure axis label may have been misleading to the reader and Reviewer #3, hiding this relevant finding within the paper. To fix this, we have updated the text in the specificity portion of the Results to include:

"In addition, the nickase construct only increased base editing signal at the intended on-target site (**supplementary figure 8a**), implying such boosts to base editing signal can be very specific with the ZF-CBE-nickase architecture used in this study."

We have also updated the y-axis in **supplementary figure 8a** from "% A/T change with nickase ZFP" to "% A/T change with addition of the nickase ZFP".

Regarding an analysis of off-targets caused by the nickase alone, we would like to point Reviewer 3 to our response to Reviewer 1 regarding long-term nickase expression in an AAV therapy application of the ZF-CBE-nickases.

4. The title of the manuscript is too general. Nickase and/or newly identified TDD should be included.

We adjusted the title to "A compact zinc finger architecture utilizing novel cytidine deaminases for highly efficient base editing in human cells" as suggested by Reviewer 3.

Reviewers' Comments:

Reviewer #1:

Remarks to the Author:

I am satisfied with the response of the authors

Reviewer #2:

Remarks to the Author:

The authors have adequately addressed some of the comments and suggestions made. Clarification has been given for amendments that fell outside the scope of their study. Packaging of all components in a single AAV vector system would still have been nice to have seen and would have considerably strengthened the manuscript, but the work-up required to show this would be significant.

I have no further comments.

Reviewer #3:

Remarks to the Author:

The authors have addressed most of my concerns. I support the publication. When studying off-target effects for ZFP-CBE-nickase, the authors used a panel of off-target sites that are derived from ZFP-CBE assay. Can authors comment on this? Is it possible ZFP-CBE-nickase may cause off-target editing at different sites?

We were happy to hear that all three reviewers support the publication. We would like to comment on a new remark by reviewer #3. Reviewer #3 asked:

“When studying off-target effects for ZFP-CBE-nickase, the authors used a panel of off-target sites that are derived from ZFP-CBE assay. Can authors comment on this? Is it possible ZFP-CBE-nickase may cause off-target editing at different sites?”

We identified ZF-CBE off-targets using a biochemical assay and purified genomic DNA. The nickase portion of our base editors requires the cellular DNA repair machinery to impact on- and off-target base editing efficiency levels. Consequently, including a nickase in such a base editing off-target assays would not be likely to identify additional off-target sites. However, we discussed in our revised manuscript how nickase off-target sites can be identified if relevant:

“The level of off-target nicking was not measured in this study, but could be investigated for therapeutic applications by reversing the D450N FokI mutation and co-transfecting a dsDNA oligonucleotide to identify the resulting dsDNA breaks (Miller et al., 2019).”

Please let us know if you think that this doesn't sufficiently address reviewer 3's concern.

We attached all requested documents to this submission.

Thanks,
Jeff